# Assessing Risky Driving Behaviours of Chinese Drivers Aged 55–65 Years: Adaptation of the Road Traffic Behaviours Questionnaire and Its Associations with Personality Traits and Social Desirability

**DOI:** 10.3390/bs15121720

**Published:** 2025-12-12

**Authors:** Weiyi Chen, Liang Cheng, Long Sun

**Affiliations:** School of Psychology, Liaoning Normal University, Dalian 116021, China; 15842427243@163.com (W.C.); chengliangsea@163.com (L.C.)

**Keywords:** risky driving behaviour, personality traits, social desirability

## Abstract

Although many instruments assess driving behaviour, the validity of the Chinese versions of these tools in assessing the driving behaviours of drivers aged over 60 years remains largely unexamined. Additionally, the number of Chinese drivers over 55 obtaining licenses continues to rise, yet links between risky driving and crashes in this group are underexplored. To address these gaps, the Road Traffic Behaviours Questionnaire (RTBQ) was adapted to 320 drivers aged 55–65 years. Participants completed questionnaires assessing personality traits, social desirability, and driving behaviour. The finalized Chinese version of the RTBQ contains 13 questions and demonstrates excellent reliability. Significant associations among the RTBQ score, personality traits, social desirability and aggressive and prosocial driving behaviours suggest that its convergent and discriminant validity are acceptable. Finally, drivers with previous traffic accidents scored significantly higher on the RTBQ than those without traffic accidents, indicating its known-group validity is satisfactory. The RTBQ score can also predict traffic accidents in the following 6 months. The reliable and validated RTBQ has the potential to be used for subsequent research on Chinese drivers aged 55–65 years and provides empirical evidence for traffic safety policy-making in China.

## 1. Introduction

During the first quarter of 2025, the number of elderly drivers aged 60 years and above in China exceeded 85 million, accounting for 17.2% of all drivers in China. Additionally, the number of serious traffic accidents involving these drivers accounted for more than 20% of all accidents ([23]). With the accelerating process of societal ageing in China, the number of individuals aged over 60 years who are obtaining driver’s licenses is increasing every year, and their risky driving behaviours have become an important cause of road traffic accidents ([5]). Owing to the decline in cognitive, behavioural and physiological functions, elderly drivers are more prone to risky behaviours such as distracted driving, abnormal speed maintenance, violations at intersections and lane-keeping errors ([27]). To our knowledge, research on risky driving behaviour and crash risk of elderly drivers in China is still relatively limited. [18] ([18]) used a driving simulator to examine the driving performance of drivers over 65 years old and reported that elderly drivers are slow to react and have delayed starts at signal-controlled intersections, and they tend to be hesitant and have reduced judgment at unsignalized intersections. However, there are still relatively few focusing on drivers aged 55–65 who are facing retirement.

### 1.1. Risky Driving Behaviour

There are many instruments that include questions or factors related to assessing drivers’ risky driving behaviour, such as the Driver Behaviour Questionnaire (DBQ, [29]), the Dula Dangerous Driving Index (DDDI, [10]) and the Prosocial and Aggressive Driving Inventory (PADI, [13]). The DBQ is a widely used scale that provides a comprehensive assessment of driving, including intentional violations (e.g., speeding) and unintentional behaviours (e.g., errors) ([29]). [24] ([24]) reported that compared with middle-aged drivers, elderly drivers aged 65 years and above scored significantly lower in terms of violations and errors. [6] ([6]) found that compared with young drivers, elderly drivers aged 65 years and above scored higher in terms of attention lapses and unintentional errors, reflecting an increase in operational mistakes due to cognitive ageing.

The DDDI aims to evaluate risky driving through four subscales, namely, risky driving, negative cognitive/emotional driving, aggressive driving, and drunk driving, and it is suitable for drivers aged 18–72 years ([10]). [34] ([34]) reported that the scores of the four DDDI factors were significantly correlated with self-reported accident histories (age range: 17–78 years). The PADI is primarily used to measure safe (prosocial) and unsafe (aggressive) driving behaviour and is suitable for drivers aged 18–72 years ([13]). [13] ([13]) reported that the prevalence of aggressive driving among drivers aged 18–25 years was significantly higher than that among drivers over 45 years, while the prevalence of prosocial driving increased linearly with age. The Chinese version of the DDDI is suitable for drivers aged 18–61 years ([28]), while the PADI is suitable for drivers aged 24–44 years ([31]). To our knowledge, the validity of the two scales for assessing the driving behaviour of Chinese drivers aged 60 years and above still needs more investigation.

[2] ([2]) developed the Road Traffic Behaviours Questionnaire (RTBQ) and validated it in a Polish sample of drivers aged 18–77 years. Compared with the DBQ and the DDDI, the RTBQ is a single-dimensional questionnaire and contains 16 questions. In practical use, the RTBQ can take the total score or each question as the dependent variable to assess the risk tendency of drivers aged 55–65 years. Moreover, the 16 questions correspond to the most frequently reported dangerous driving behaviours among Chinese drivers of varying ages ([1]). Additionally, the education level of most Chinese drivers aged 55–65 years is primary and middle school (in contrast to young drivers), which indicates that a short and unidimensional questionnaire could be more attractive and convenient for this driver group in terms of large-scale testing or online surveys for personal purposes. Given the discussion above, this study aims to translate and adapt the RTBQ to Chinese drivers aged 55–65 years.

### 1.2. Risky Driving Behaviour and Personality Traits

Many studies have shown that personality traits can predict driving behaviour (e.g., [19]). The relationships between altruism and risky behaviours were confirmed across studies ([9]; [19]). For instance, via the use of a sample of 1286 Italian drivers aged 60 to 90 years, [20] ([20]) reported that altruism was negatively correlated with violations and accident records. Although studies have shown that anger can positively capture risky driving behaviour ([12]; [35]), [37] ([37]) reported that higher trait anger was linked to fewer traffic crashes and fines.

Sensation seeking can also predict risky driving behaviour ([8]). However, [19] ([19]) reported that sensation seeking was not significantly associated with unintentional driving errors/lapses among drivers aged over 65 years. [2] ([2]) also failed to find a significant association between sensation seeking and risky driving behaviour, as measured via the RTBQ. Although normlessness can predict driving behaviour (e.g., [9]), some studies have reported that normlessness cannot significantly predict self-reported risky behaviours or the number of violations during road tests among elderly drivers (e.g., [3]). Given these inconsistent results, this study selected four personality traits (sensation seeking, trait anger, altruism, and normlessness) and further explored the influence of these personality traits on the risky driving behaviour of drivers aged 55–65 years.

### 1.3. Risky Driving Behaviour and Social Desirability

Social desirability bias is defined as the tendency to provide socially accepted and favourable answers ([26]). It is often regarded as a confounding variable that may mask true driving behaviours ([17]; [36]). According to the two-factor model of social desirability bias proposed by [26] ([26]), this bias includes self-deception (i.e., respondents believing in their exaggerated positive responses in their hearts and responding honestly correspondingly) and impression management (i.e., the conscious efforts of respondents to shape a good impression of themselves by providing positive responses).

Studies have shown that impression management is negatively correlated with the number of self-reported traffic accidents and driving aggression ([17]) and aggressive and ordinary violations ([36]), whereas drivers’ self-deception is positively related to their sense of control in traffic ([17]) and positive behaviours ([36]). [1] ([1]) reported that social desirability is negatively correlated with the reckless driving behaviour of young Chinese drivers aged 18–25 years and that it plays a partial mediating role in the relationship between reckless driving and previous traffic violations. Given that China is a country with a collectivist culture and that people tend to value others’ comments and opinions, this study examined the relationships between the risky driving behaviour of drivers aged 55–65 years and traffic accidents while controlling for social desirability.

### 1.4. Risky Driving Behaviour, Demographic Factors, and Traffic Accidents

Demographic factors also influence risky driving behaviour, yet the findings of prior studies have been inconsistent in terms of the gender differences in risky driving behaviour. Studies have shown that men violate traffic rules more often and accumulate more tickets than women (e.g., [7]), whereas other studies have revealed no such difference ([33]). [2] ([2]) reported significant gender differences in the RTBQ among the age group of 30–40 years, but no difference was found in drivers aged 18–29 years, 30–39 years, 40–49 years and those over 50 years. Hence, the gender and age-related differences in the RTBQ need to be further investigated.

As people age, although the number of active violations of elderly drivers decreases overall ([4]), those who crash still display elevated risky behaviours ([30]). The perception–response delays linked to physical and cognitive decline are thought to increase collision risk ([11]), but [22] ([22]) reported no difference in near-crash rates between drivers aged 70–86 years and 18–21 years. In this regard, this study further examines the differences between drivers aged 55–65 years who had and did not have a record of traffic accidents.

### 1.5. Objectives of This Study

Given that the validity of the Chinese versions of measures, such as the PADI and the DDDI in assessing the driving behaviours of drivers aged over 60 years remains largely unexamined, the first objective of this study is to translate and validate the RTBQ to Chinese drivers aged 55–65 years to identify different levels of risky driving behaviours.

The second objective is to explore the associations between the RTBQ score, prosocial and aggressive driving behaviour, personality traits, and social desirability to assess its convergent and discriminant validity of the RTBQ. Given the significant associations between risky driving behaviours and personality traits ([20]; [35]), we predict that RTBQ score is positively associated with aggressive driving behaviour, trait anger, sensation seeking and normlessness, and is negatively associated with prosocial driving behaviour and altruism. Additionally, China is a country with a collectivist culture and that people tend to value others’ comments and opinions ([32]), hence the relationship between social desirability and risky driving behaviours was also explored to assess its discriminant validity. We predicted that higher impression management would relate to lower risky driving, whereas higher self-deception would relate to higher risky driving.

The third objective is to explore the relationships between risky driving behaviour and both prospective and retrospective accident data. As the number of Chinese drivers over age 55 obtaining licenses increases annually, clarifying these associations can inform driver training and licensing. The known-group validity of the RTBQ was examined by comparing the RTBQ score between drivers with and without previous traffic accidents after controlling for social desirability. In line with the findings of previous studies showing that a high level of risky driving behaviour is positively correlated with self-reported accidents ([2]; [21]), we predict that drivers with traffic accidents scored higher on the RTBQ than those without such histories. Predictive validity of the RTBQ was assessed by tracking crashes during the six months following baseline; we predicted that higher RTBQ scores would be associated with more crashes during follow-up.

Considering that the education level of most Chinese drivers aged 55–65 years is primary and middle school, the fourth objective is to explore whether risky driving behaviour differed across demographic factors such as years of education and gender.

## 2. Methods

### 2.1. Participants

A total of 320 drivers aged 55 to 65 years were recruited; 21 drivers refused to take the test after they understood the purpose of the study. The inclusion criteria were (a) being aged between 55 and 65 years and (b) holding a driver’s licence. After incomplete answers were removed, 279 valid responses were obtained. The final sample included 212 men and 67 women, with an average age of 59.67 years (*SD* = 3.03). Years of driving experience ranged from 0.5 to 13 years (*M* = 3.81, *SD* = 3.78). Forty-seven percent of the participants were newly licenced drivers whose driving experience was less than 1 year, and 53% had more than one year of driving experience. With respect to traffic accidents, 255 drivers did not have an accident last year, and 24 drivers had at least one accident last year. All traffic accidents in the past 12 months were recorded by the traffic police bureau, and drivers took full or main responsibility for their accidents.

Six months after the initial recruitment, the accident data of the participants were obtained through telephone interview. However, the data of 15 drivers were missing due to number change or they simply refused to provide their accident data. The sample consisted of 200 men and 64 women, with an average age of 59.63 years (*SD* = 3.04). The driving experience ranged from 1 to 13.5 years (*M* = 3.91, *SD* = 3.83). During the six-month follow-up, 250 drivers reported no crashes and 14 reported one crash. Detailed information of the participants is presented in Table 1.

### 2.2. Procedures and Measures

A convenient sample was used in the present study. A questionnaire survey was conducted online via the Wenjuanxing platform from March 6 to 31, 2025. The participants were required to complete the questionnaires described in this section within 30 min. Each participant received 10 yuan upon completion.

#### 2.2.1. Road Traffic Behaviour Questionnaire

The RTBQ consists of 16 questions, which are rated on a 4-point Likert scale ranging from 0 (never) to 3 (always). The scale was translated by two qualified researchers following a translation/back-translation procedure. To ensure cultural appropriateness, the questions were revised to reflect China’s right-hand traffic system, the prevalence of two-wheelers and non-motorized vehicles in mixed traffic, and local norms such as yielding to pedestrians at zebra crossings. In this regard, the question honking at drivers who drive too slowly was changed to honking at other road users (drivers, e-bike cyclists, cyclists and motorcyclists). Then, one traffic psychology safety expert (male, 34 years old, with 8.5 years of driving experience) and one driver (male, 60 years old, with 1.25 years of driving experience) were invited to rate the questions, and the results revealed high degree of consistencies in accuracy and frequency across 15 questions (using interrater reliability). They suggested that one question (racing with another car) is rarely reported among drivers aged 55–65 years. In this regard, this question was changed to speaking or answering on a mobile phone (whether hand-free or not) while driving. A small group of 30 drivers aged 55–65 years was then asked to complete the RTBQ, and they suggested revision to one question. Hence, the question being caught by police speed radar was changed to drive slightly over speed limit due to inattention. After this revision, the 16 questions were retained for further analysis.

#### 2.2.2. Prosocial and Aggressive Driving Inventory (PADI)

This scale comprises 28 questions and two factors ([13]), namely, prosocial behaviour (e.g., decrease speed to accommodate poor weather conditions) and aggressive behaviour (e.g., make rude gestures at other drivers when they do something I don’t like), and it was translated into Chinese in 2018 ([31]). Questions are rated on a 6-point Likert scale ranging from 1 (never) to 6 (always). In this study, the internal reliability of prosocial driving behaviour was 0.93, and that of aggressive driving behaviour was 0.87.

#### 2.2.3. Personality Trait Scale

The personality scale consists of 34 questions and four factors: trait anger (10 questions, I often get angry at the way people treat me, Cronbach α = 0.860), sensation seeking (10 questions, I like to do things that are a little dangerous, α = 0.852), altruism (10 questions, I try to help others who are in difficulty, α = 0.853), and normlessness (4 questions, Laws are made to be broken, α = 0.722). This scale is widely used in road safety research in China ([35]). Anger refers to the tendency of people to view events around them as sources of frustration or annoyance. Sensation seeking reflects an individual’s active embrace of risks to experience the thrill of driving. Altruism refers to a cooperative and kind-hearted inclination, always caring for others. Normlessness refers to an individual firmly believing that only through unapproved means can set goals be achieved.

#### 2.2.4. Driver Social Desirability Scale (DSDS)

The DSDS comprises 12 questions and two factors ([17]): impression management (e.g., I have never exceeded the speed limit) and self-deception (e.g., I always anticipate the actions of other road users). Questions are rated on a 7-point Likert scale ranging from 1 (never) to 7 (always). The validity and reliability of the Chinese version of the DSDS were reported elsewhere (under review). The results of confirmatory factor analysis indicated that the model fit of the scale was satisfactory, CMIN = 124.15, *df* = 53, CMIN/*df* = 2.34, GFI = 0.96, TLI = 0.97, CFI = 0.98, RMSEA = 0.054. The reliability (Cronbach’s α) for driver impression management and self-deception were 0.89 and 0.91, respectively.

### 2.3. Statistical Analysis

All statistical analyses were conducted using SPSS version 27.0 and AMOS version 26.0. The analysis procedures were structured according to the specific research objectives of this study.

For the first objective, we examined item properties (means, standard deviations, skewness, kurtosis), followed by an exploratory factor analysis (oblimin rotation) and a confirmatory factor analysis (maximum likelihood) to evaluate the RTBQ factor structure. Internal consistency reliability (Cronbach’s alpha), composite reliability (CR), and the average variance extracted (AVE) were computed to assess the reliability and convergent validity of the scale. For CFA, adopting the criteria suggested by [14] ([14]), good fit index (GFI), comparative fit index (CFI) and Tucker–Lewis index (TLI) values greater than 0.9 and an approximate root mean square error (RMSEA) value less than 0.08 indicated that the fitness index was acceptable.

For the second objective, Pearson correlation analyses were conducted to examine the relationships between the RTBQ score, personality traits, the DSDS and the PADI to evaluate discriminant and convergent validity. A hierarchical regression analysis was performed to determine whether personality traits and socially desirable response predicted risky driving behaviour after demographic variables were controlled. A two-step cluster analysis was also conducted to identify subtypes of drivers based on the studied variables.

For the third objective, one-way analysis of variance (ANOVA) was conducted to compare the RTBQ score, prosocial and aggressive behaviours between drivers with and without a previous history of traffic accidents to assess the known-group validity of the RTBQ. The differences in prospective accident data of the drivers identified through Cluster analysis were also compared to assess the predictive validity of the RTBQ.

For the fourth objective, one-way ANOVA was used to examine gender differences in the RTBQ score, and Pearson correlation analyses were conducted to examine the relationships between age, years of education, years of experience, and the RTBQ score.

## 3. Results

### 3.1. Item Analysis

The means, standard deviations, skewness and kurtosis of the 16 questions were analysed (see Table 2). [16] ([16]) suggested that the skewness and kurtosis values of questions should be within the ranges of ±3 and ±7, respectively, indicating normality. All the questions exhibited skewness values between 0.25 and 0.62 and kurtosis values between −1.11 and −0.74. Hence, they were retained for the EFA.

### 3.2. EFA and CFA

The original RTBQ questions was submitted to an EFA through principal component analysis with oblique rotation, and the results (see Table 2) revealed two factors (eigenvalue >1) that explained 49.11% of the variance, KMO *=* 0.942, Bartlett’s spherical test *=* 1659.15, *p* < 0.01. However, three questions became a separate factor (the internal consistency reliability was 0.48) and were deleted. The remaining 13 questions were submitted to a second EFA through principal component analysis with oblique rotation. The results revealed one factor that explained 47.99% of the variance, KMO *=* 0.947, Bartlett’s sphericity test *=* 1476.85, *p* < 0.01.

CFA was conducted via the maximum likelihood estimation method to examine the model fitness of the RTBQ via AMOS. According to the cut-off values proposed by ([14]), the model fit was satisfactory, CMIN *=* 156.67, *df =* 104, CMIN/*df =* 1.506, GFI = 0.935, CFI = 0.967, RMSEA = 0.043.

### 3.3. Reliability and Validity of the RTBQ

Reliability analysis indicated that the reliability of the RTBQ was 0.901, AVE = 0.61, CR = 0.75. Pearson correlation was used to examine the correlations between the RTBQ score, PADI, personality traits, and DSDS factors. The results are shown in Table 3.

Table 3 shows that the RTBQ score was positively associated with aggressive driving behaviours, trait anger, sensation seeking, normlessness, self-deception and impression management, and negatively associated with prosocial driving behaviour and altruism. The results indicate that the convergent and discriminant validity of the RTBQ are acceptable. The Spearman correlation results show that the number of traffic accidents is positively correlated with the RTBQ score (*r* = 0.16, *p* < 0.01), and the number of traffic accidents in the following six months is still positively correlated with the RTBQ score (*r* = 0.14, *p* < 0.05).

### 3.4. Regression Analysis and Cluster Analysis

To explore the predictive factors of the RTBQ, the total RTBQ score was used as the dependent variable, personality traits and social desirability bias were used as independent variables, and a hierarchical regression analysis was conducted after controlling for demographic factors (see Table 4). Table 4 shows that trait anger, altruism, and self-deception can significantly predict the RTBQ score.

A two-step cluster analysis was conducted to identify subtypes of drivers (see Table 5). The analysis was based on scores from personality traits, DSDS factors and the RTBQ score, using the squared log-likelihood distance measure. The analysis identified two clusters as the best fit for the data based on Schwarz’s Bayesian Criterion score and the largest ratio of log-likelihood distance measures. Table 5 shows that drivers (53.4%) in Cluster 1 scored lower in impression management and altruism, while they scored higher in trait anger, normlessness, sensation seeking, self-deception and the RTBQ. The reversed trend was found among drivers (46.6%) in Cluster 2.

The result of one-way ANOVA shows that drivers in Cluster 1 (*M* = 0.14, *SD* = 0.14) had more traffic accidents than drivers in Cluster 2 (*M* = 0.06, *SD* = 0.13), *F*(1, 277) = 4.04, *p* = 0.045, *η*_p_^2^ = 0.014. Additionally, in the driving of the following 6 months after the first data collection, drivers in Cluster 1 (*M* = 0.08, *SD* = 0.27) still had more traffic accidents than drivers in Cluster 2 (*M* = 0.02, *SD* = 0.15), *F*(1, 262) = 3.79, *p* = 0.05, *η*_p_^2^ = 0.014.

### 3.5. The RTBQ and Traffic Accidents

One-way ANOVAs were conducted to compare the differences in the RTBQ total score, prosocial behaviour and aggressive behaviour between drivers with and without accidents. The results are provided in Table 6.

Table 6 shows that drivers with traffic accidents scored higher on the RTBQ than did those without accidents (see Figure 1). The results of analysis of covariance (ANCOVA) confirmed this trend when social desirability was used as covariate. No significant differences in prosocial or aggressive behaviour were found between drivers with accidents and without accidents.

PROCESS Model 4 was used to examine whether social desirability mediated the link between the RTBQ score and self-reported previous traffic accidents. No mediating effect of self-deception or impression management was found in this study.

### 3.6. RTBQ and Demographic Variables

The results of one-way ANOVA (see Figure 1) reveal that compared with women, men scored higher on the RTBQ, *F*(1, 277) = 5.01, *p* = 0.026. Although the score for drivers who just have obtained their driver’s license (*M* = 1.31, *SD* = 0.67) was higher than that of those with more than one year of driving experience (*M* = 1.24, *SD* = 0.72), the difference was not significant, *F*(1, 277) = 0.661, *p* = 0.417. Pearson correlations revealed that the RTBQ score was negatively correlated with years of driving experience (*r* = −0.18, *p* < 0.01) and years of education (*r* = −0.12, *p* < 0.05). No significant correlation was found between the RTBQ score and age.

## 4. Discussion

The main purpose of this study is to translate and adapt the RTBQ to Chinese drivers aged 55–65 years by incorporating Chinese traffic culture and customs. The findings show that the Chinese version of the RTBQ exhibits acceptable reliability and validity.

Our first contribution is the development and validation of a brief measure (RTBQ) of risky driving for Chinese drivers aged 55–65 years, tailored to China’s rapid population aging and the surge in late-life licensure. The final Chinese version of the RTBQ contains 13 questions with satisfactory reliability. Compared with the original version ([2]), this study removed three questions from the RTBQ (e.g., not wearing a seat belt) because they were separated into a factor with a Cronbach’s alpha lower than 0.7. The remaining 13 questions were revised to ensure cultural sensitivity on the basis of the suggestions of experts in traffic psychology and drivers in the same age range. Both exploratory factor analysis and confirmatory factor analysis verified the single-factor structure of the scale.

Convergent and discriminant validity were supported by significant associations between RTBQ scores and personality traits, as well as with established driving measures (i.e., PADI and DSDS). The RTBQ was positively correlated with aggressive behaviour and trait anger but was negatively correlated with prosocial behaviour and altruism, replicating the results of some previous studies ([25]; [31]; [35]). Research has shown that drivers aged 60–90 years with high scores on altruism are more rule-abiding ([20]), resulting in more safe driving behaviours and less risky driving behaviours ([7]). In line with the findings of some previous Chinese studies ([12]; [35]), this study revealed that trait anger can significantly and positively predict risky driving behaviour, as measured by the RTBQ. In a context of steadily increasing traffic density, Chinese drivers may encounter more anger-provoking situations.

A notable finding is that self-deception, rather than impression management, can significantly and positively predict the RTBQ scores. [17] ([17]) reported that self-deception is more related to overestimating one’s own abilities than to underestimating the danger of one’s own behaviour. Given that 47% of the participants in this study were newly licenced drivers, it is possible that some of them might not have accurately assessed their driving abilities. This study revealed a negative association between impression management and the RTBQ score, although it could not significantly predict risky driving behaviour. Notably, [36] ([36]) reported that impression management can predict aggressive violations and ordinary violations among Turkish drivers aged 19–59 years. This pattern aligns with Chinese collectivist sociocultural context. Drivers motivated to make a good impression are likely to adopt safer behaviours to obtain social approval ([32]).

Another contribution of this study is that both the known-group validity and predictive validity of the RTBQ were confirmed by examining the relationships between the RTBQ score and both prospective and retrospective accident data. This study revealed that drivers with previous accidents scored higher on the RTBQ than did drivers without accidents, supporting the known-group validity of the RTBQ. Notably, no significant differences in either prosocial or aggressive driving behaviours emerged between drivers with and without previous traffic accidents, indicating the necessity of re-examination of the validity of existing measure in studies focusing on drivers aged 55–65 years in China. The results of the cluster analysis showed that the high-risk drivers identified in Cluster 1 scored lower on altruism and impression management, while higher scores in trait anger, normlessness, sensation seeking, self-deception, and the score of RTBQ, indicating a higher tendency towards being involved in traffic accidents. Drivers in Cluster 2 (low-risk) have the reversed trend. Moreover, drivers in Cluster 1 had more traffic accidents than did those in the Cluster 2. The findings align with [2] ([2]), who emphasized that the RTBQ is effective in identifying driver groups with varying risk.

This study is the first to show that RTBQ scores predict crash risk over a six-month period, even with a low base rate of crashes. Higher RTBQ scores were associated with more crashes six months later, and the high-risk cluster (Cluster 1) reported more traffic accidents than the lower-risk cluster (Cluster 2). These results position the RTBQ as a brief, context-specific tool with practical utility for risk stratification, longitudinal research, and targeted prevention among older Chinese drivers.

Finally, this study found that risky driving behaviour of drivers aged 55–65 years differed across demographic factors such as years of education and gender. This study addressed the research gap that focused on drivers aged less than 55 years or those aged over 65 years. As many Chinese adults in this age range approach retirement and late-life licensing continues to grow, documenting their driving patterns has clear preventive value. Specifically, this study found that men drivers scored higher on the RTBQ than women. This finding diverges from [2] ([2]), which revealed no gender differences among drivers older than 50 years. Further research is warranted to examine gender-related risk specifically in drivers aged 55 years and older. Although the education level of the participants in this study is mainly primary and secondary school education, this study still found that additional years of education could be a protective factor in reducing risky driving behaviour. Notably, although 47% of the participants had held a licence for less than one year, yet their RTBQ scores did not differ significantly from those of the more experienced drivers. The absence of an RTBQ difference probably stems from newly licenced drivers may still adhere closely to driving-school rules and have not consolidated stable, personal driving styles.

## 5. Implications

The Chinese version of the RTBQ shows acceptable reliability and validity, providing a short, user-friendly, and brief tool for measuring the risky driving behaviour of drivers aged 55–65 years. The RTBQ can also be used to evaluate the effects of driver training programmes targeting candidates who are applying their driving licences in this age range. Furthermore, by integrating demographics, personality traits, and social desirability factors, this study provides a model in which these variables together shape the driving behaviour of drivers aged 55–65 years, providing insights for subsequent intervention measures. For instance, drivers who scored higher on trait anger and self-deception could choose to take an online supplementary test, including measures such as the RTBQ, during the theory part of the driver’s licence test to help correctly assess their driving abilities and risk tendencies.

## 6. Limitations

This study exhibits limitations. The sample size is small, and the drivers are mainly recruited from urban areas. In the future, a large sample of drivers with varying levels of driving experience and age should be recruited to further verify the results of this study. The second limitation is that this study relies solely on self-reported data; future research should aim to integrate objective data, such as in-vehicle monitoring data or driving simulator records ([15]), to further refine the psychometric properties of the RTBQ. Third, this study did not cover drivers who were older than 65 years. Although the number of drivers aged above 65 years is relatively small and they might not be easily recruited, future studies should examine the validity of the RTBQ in drivers whose age are above 65 years.

## 7. Conclusions

This study provides a brief and valid scale in assessing risky driving behaviour of Chinese drivers aged 55–65 years by adapting the RTBQ to Chinese drivers in this age range. The Chinese version of the RTBQ demonstrates robust reliability, convergent and discriminant validity and known-group validity. More importantly, this study unpacks why certain personality traits (altruism, trait anger) and social-desirability components (self-deception) predict risky driving behaviour in this age group, and identifies a high-risk driver group that are more likely to be involved in traffic accidents. Additionally, the predictive validity of the RTBQ in predicting crash risk in the driving of the following 6 months was confirmed. Overall, the results offer an evidence-based, culturally sensitive tool in driver assessment and training targeting at Chinese drivers aged 55–65 years.

## Figures and Tables

**Figure 1 behavsci-15-01720-f001:**
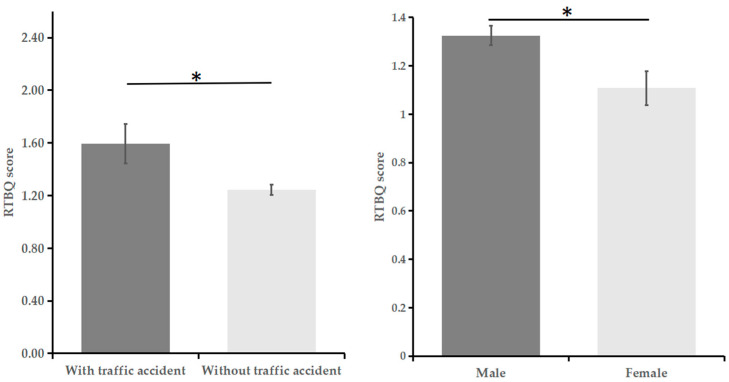
Differences in the RTBQ score by traffic accidents (**left panel**) and gender (**right panel**). * *p* < 0.05.

**Table 1 behavsci-15-01720-t001:** Participant demographics.

Variables	First Round of Data Collection		Second Round of Data Collection	
	*n*	Percent (%)	*n*	Percent (%)
Age groups by gender				
55–60 years old				
Man	122	43.7	115	43.6
Woman	41	14.7	40	15.2
61–65 years old				
Man	90	32.3	85	32.2
Woman	26	9.3	24	9.1
Years of Education				
6 years	82	29.4	78	29.5
9 years	133	47.7	124	47
12 years	64	22.9	62	23.5
Driving experience				
≤1 year	131	46.9	121	45.7
1 year–3 years	38	13.7	36	13.7
3 years–5 years	30	10.7	28	10.6
>5 years	80	28.7	79	30
Driving frequency				
Driving more than 3 times a week	74	29.7	75	28.4
Driving 1 to 2 times a week	83	26.5	71	26.9
Driving 3 to 4 times a month	88	31.5	85	32.2
Driving occasionally	34	12.2	33	12.5
Accident last year				
0 accident	255	91.4	250	94.7
1 accident	17	6.1	14	5.3
2 accidents	7	2.5	0	0

**Table 2 behavsci-15-01720-t002:** Factor loadings of the RTBQ (*n* = 279).

Questions	Factor Loadings	M	SD	Skewness	Kurtosis	Question-Scale Correlation
1. Driving very close to the vehicle in front of you	0.644	2.36	1.02	0.317	−0.987	0.741 **
2. Forcing the right of way	0.639	2.34	1.00	0.250	−0.988	0.673 **
3. Overtaking on a continuous line	0.617	2.23	0.95	0.405	−0.746	0.678 **
4. Overtaking another car in conditions of very low visibility	0.630	2.31	0.99	0.330	−0.897	0.675 **
5. Driving at high speed in poor weather conditions (i.e., fog, rainfall, sleet)	0.746	2.25	1.01	0.372	−0.929	0.717 **
6. Driving a car without lights on	0.732	2.25	1.02	0.303	−1.028	0.698 **
7. Not reducing the speed when approaching a railway crossing	0.727	2.35	1.06	0.313	−1.113	0.707 **
8. Changing traffic lane without signalling it to other drivers	0.510	2.27	0.99	0.325	−0.936	0.644 **
9. Speaking or answering on a mobile phone (whether hand-free or not) while driving	0.583	2.25	1.04	0.345	−1.056	0.663 **
10. Cutting in on another car	0.668	2.21	0.97	0.408	−0.804	0.672 **
11. Drive slightly over speed limit due to inattention	0.690	2.22	1.02	0.430	−0.917	0.652 **
12. Honking at other road users (drivers, e-bike cyclists, cyclists and motorcyclists)	0.588	2.27	1.05	0.343	−1.061	0.702 **
13. Texting while driving	0.686	2.24	0.97	0.377	−0.817	0.625 **

** *p* < 0.01.

**Table 3 behavsci-15-01720-t003:** Correlations among the studied variables (*n* = 279).

Variables	1	2	3	4	5	6	7	8	9
RTBQ (1)	1	−0.20 **	0.28 **	0.39 **	0.30 **	−0.43 **	0.24 **	−0.22 **	0.34 **
Prosocial driving behaviour (2)		1	−0.46 **	−0.19 **	−0.34 **	0.33 **	−0.29 **	0.33 **	−0.26 **
Aggressive driving behaviour (3)			1	0.14 *	0.20 **	−0.38 **	0.30 **	−0.27 **	0.31 **
Trait anger (4)				1	0.47 **	−0.54 **	0.43 **	−0.45 **	0.45 **
Sensation seeking (5)					1	−0.45 **	0.45 **	−0.44 **	0.34 **
Altruism (6)						1	−0.49 **	0.49 **	−0.44 **
Normlessness (7)							1	−0.45 **	0.35 **
Impression management (8)								1	−0.40 **
Self-deception (9)									1

* *p* < 0.05; ** *p* < 0.01.

**Table 4 behavsci-15-01720-t004:** Hierarchical regression coefficients for predicting risky driving behaviour (*n* = 279).

Variables		Beta	*t*	Δ*R*^2^
Step 1	Gender	−0.12	−2.24 *	0.055 **
	Age	−0.06	−1.07	
	Driving experience	−0.13	−2.33 *	
	Years of education	−0.12	−2.27 *	
	Driving frequency	−0.03	−0.55	
Step 2	Trait anger	0.18	2.65 *	0.17 **
	Sensation seeking	0.06	0.94	
	Altruism	−0.26	−3.59 **	
	Normlessness	−0.02	−0.31	
Step 3	Impression management	0.09	1.34	0.10 **
	Self-deception	0.13	2.15 *	

* *p* < 0.05; ** *p* < 0.01.

**Table 5 behavsci-15-01720-t005:** Cluster differences in personality traits, DSDS factors and RTBQ score (*n* = 279).

Variables	Cluster 1 (*n* = 149)		Cluster 2 (*n* = 130)		Statistics	
	*M*	*SD*	*M*	*SD*	*F*(1, 277)	*η_p_* ^2^
Personality traits						
Trait anger	3.11	0.61	2.15	0.45	218.91 **	0.44
Sensation seeking	3.06	0.72	2.10	0.51	160.83	0.37
Altruism	2.69	0.56	3.79	0.47	312.21	0.53
Normlessness	3.10	0.85	2.01	0.31	186.78	0.40
DSDS factors						
Impression management	3.87	1.30	5.51	0.63	170.71	0.38
Self-deception	4.59	1.20	2.92	1.50	105.69	0.27
RTBQ score	1.64	0.55	0.86	0.60	125.96	0.31

** *p* < 0.01.

**Table 6 behavsci-15-01720-t006:** Differences in the RTBQ, driving behaviours by traffic accidents (*n* = 279).

Variables	With Accidents (*n* = 24)		Without Accidents (*n* = 255)		*F*(1, 277)	*F*(1, 272)
	*M*	*SD*	*M*	*SD*		
Average RTBQ score	1.59	0.74	1.24	0.68	5.72 *	5.85 *
Prosocial driving behaviour	3.20	0.82	3.30	0.83	0.34	0.51
Aggressive driving behaviour	2.81	0.84	2.79	0.80	0.01	0.01

* *p* < 0.05.

## Data Availability

The original data presented in the study are openly available in [FigShare] at [https://doi.org/10.6084/m9.figshare.30343036].

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
