# Peer review of "Assessing Risky Driving Behaviours of Chinese Drivers Aged 55–65 Years: Adaptation of the Road Traffic Behaviours Questionnaire and Its Associations with Personality Traits and Social Desirability"

_behavsci, 2025, doi:10.3390/bs15121720_

Round 1

Reviewer 1 Report

Comments and Suggestions for Authors

This is an interesting paper which is trying to study the driver behavior through a sel-reported questionnaire. The authors did a very good job defining the research scope, background, and methodology, which helped put the reader into perspective.

The literature needs to be expanded to explore more recent references and to include survey validation efforts in China such as “Revision of the driver behavior questionnaire for bus drivers in China based on in-vehicle monitoring data”.

The authors always mention how many items in each survey was used, which is good. However, I would suggest using questions instead of items to avoid any confusion.

Although the introduction and methodology are well established, the results section seems to be rushed by dumping all the numbers at once without even explaining what are the measures that were reported in the results such as GFI, CFI, RMSEA. These measures need to be defined and the authors should explain what these measures are used for.

The authors acknowledged that 47% of the drivers held a driving licenece recently (i.e., less than 1 year ago) and they discussed how this may impact the results. However, they should show in Table 1, the driver distribution with respect to the years of driving and not just focusing on one year cutoff. For example, it is important to show how many drivers have held a driving licence for 1-3 years, 3 – 5 years, and more than 5 years. This will be more informative to the reader and will help understand the results better.

Author Response

A response letter was uploaded.

Reviewer 2 Report

Comments and Suggestions for Authors

Dear Authors,

This review highlights the limited validation of Chinese versions of driving behavior assessment tools for drivers over 60 and examines the adaptation of the RTBQ for Chinese drivers aged 55–65. The findings demonstrate the RTBQ's strong reliability and validity, emphasizing its potential for research and traffic safety policies in China.

The work and presentation are solid; the findings and responses obtained for the research questions are appropriate. The methods employed and the adaptation of the RTBQ for the focus group are correct.

Minor note for consideration:

  • Some visualization is needed to better represent results at the end of the Results section or in the discussion section.
  • A table summarizing key findings (metrically or descriptively) would improve the understandability of the results. 
  • Table formatting is inconsistent; please use the template.
  • The reference list and citations do not comply with MDPI rules.

If the well-reached results are better and transparently presented, the paper can be published.

Author Response

A response letter was uploaded.
